# Growth State-Dependent Activation of eNOS in Response to DHA: Involvement of p38 MAPK

**DOI:** 10.3390/ijms24098346

**Published:** 2023-05-06

**Authors:** Shiqi Huang, Carla G. Taylor, Peter Zahradka

**Affiliations:** 1Department of Food and Human Nutritional Sciences, University of Manitoba, Winnipeg, MB R3T 2N2, Canada; 2Canadian Centre for Agri-Food Research in Health and Medicine, St. Boniface Hospital Albrechtsen Research Centre, Winnipeg, MB R2H 2A6, Canada; 3Department of Physiology and Pathophysiology, University of Manitoba, Winnipeg, MB R3E 0W2, Canada

**Keywords:** eNOS, DHA, p38 MAPK, MSK, endothelial cells, growth state

## Abstract

Our laboratory previously reported that docosahexaenoic acid (DHA) differentially activates p38 mitogen-activated protein kinase (MAPK) in growing and quiescent human endothelial cells, which represent the dysfunctional and healthy states in vivo, respectively. Since endothelial nitric oxide synthase (eNOS) activity differs between healthy and dysfunctional endothelial cells, and p38 MAPK reportedly regulates both the activity and expression of eNOS, we hypothesized that the beneficial actions of DHA on endothelial cells are due to eNOS activation by p38 MAPK. The contribution of mitogen- and stress-activated protein kinase (MSK), a p38 MAPK substrate, was also investigated. Growing and quiescent EA.hy926 cells, prepared on Matrigel^®^-coated plates, were incubated with inhibitors of p38MAPK or MSK before adding DHA. eNOS phosphorylation and levels were quantified by Western blotting. Treatment with 20 µM DHA activated eNOS in both growth states whereas 125 µM DHA suppressed eNOS activation in growing cells. Quiescent cells had higher basal levels of eNOS than growing cells, while 125 µM DHA decreased eNOS levels in both growth states. p38 MAPK inhibition enhanced eNOS activation in quiescent cells but suppressed it in growing cells. Interestingly, 125 µM DHA counteracted these effects of p38 MAPK inhibition in both growth states. MSK was required for eNOS activation in both growth states, but it only mediated eNOS activation by DHA in quiescent cells. MSK thus affects eNOS via a pathway independent of p38MAPK. Quiescent cells were also more resistant to the apoptosis-inducing effect of 125 µM DHA compared to growing cells. The growth state-dependent regulation of p38MAPK and eNOS by DHA provides novel insight into the molecular mechanisms by which DHA influences endothelial cell function.

## 1. Introduction

Endothelial cells are responsible for controlling the passage of all substances, including nutrients, between the bloodstream and the underlying tissues. They are also critical for maintaining cardiovascular homeostasis and regulating a variety of physiological processes in our body [1]. Endothelial cells are typically quiescent in the healthy state. Damage to a vessel activates endothelial cells, which will then proliferate as part of the wound healing process [1]. These cells return to their quiescent state after the vessel wall has been repaired. However, endothelial cells can remain activated even when there is no physical damage to the vessel, and this is termed endothelial dysfunction (ED). These activated cells make the vessel susceptible to plaque formation as a consequence of reduced endothelial nitric oxide synthase (eNOS) activity, which lowers nitric oxide (NO) production [2]. Additionally, the reduction in eNOS activity enables arterial remodeling, which ultimately leads to arterial stiffness [3]. All of these processes occur due to changes in endothelial cell state, eventually progressing to atherosclerosis, the narrowing of blood vessels due to plaque formation as a result of ED [4]. These important roles of endothelial cells make them key players in atherogenesis and, thus, critical targets for therapeutic interventions designed to prevent atherosclerosis. 

eNOS is a heme-containing, calcium-dependent, and calmodulin (CaM)-modulated enzyme that catalyzes the conversion of _L_-arginine to NO and citrulline. eNOS contains 3 major domains: an N-terminal oxygenase domain, the CaM binding domain, and a C-terminal reductase domain. The activity of eNOS is heavily regulated by phosphorylation, especially at the Ser-1177 site, which is located within the reductase domain [2]. Phosphorylation at this site is believed to enhance electron transfer to the oxygenase domain that catalyzes the production of NO from L-arginine and prevents CaM dissociation when the calcium concentration is low [5]. Under certain conditions, such as an oxidative environment [3], the electron flux is uncoupled from _L_-arginine oxidation and is instead transferred to O_2_ to generate superoxide. The consequent reduction in NO bioavailability is a critical feature of ED. NO is generally regarded as the gate-keeper for endothelial function [6], and the release of NO may be used to determine the health of endothelial cells. To measure NO, however, is challenging due to its gaseous nature and very short half-life of 1–2 s. Therefore, eNOS phosphorylation (p-eNOS) status, especially at Ser-1177, is commonly used as an indicator of eNOS activity. eNOS activity is regulated at multiple tiers [3] and is also found to be affected by many dietary components, such as polyphenols [7], fructose [8], and omega-3 polyunsaturated fatty acids (n3 PUFA) [9].

Although controversies remain, it is generally agreed that n3 PUFA, particularly those from marine sources such as eicosapentaenoic acid (EPA) and docosahexaenoic acid (DHA), can reduce both plaque formation and ischemic events [10], but the underlying mechanisms by which they function still remain largely unknown. It was previously reported by our lab that DHA, but not EPA or other fatty acids, can suppress the proliferation and migration of cultured endothelial cells [11]. As well, DHA differentially activated p38 mitogen-activated protein kinase (MAPK) in sub-confluent and confluent EA.hy926 endothelial cells [12], which resemble the dysfunctional and healthy states in vivo, respectively. p38 MAPK is a member of the MAPK family that responds to extracellular stress signals including inflammation [13]. Since there is evidence that p38 MAPK can regulate eNOS activity in endothelial cells [7], it is plausible that DHA may be able to modulate the activity of eNOS via p38 MAPK. Therefore, we hypothesize that the atheroprotective effect of DHA may come from its beneficial effects on endothelial cells as a result of eNOS activation in response to stimulation of p38 MAPK signaling. In this study, we compared eNOS activation and expression in response to DHA, with or without p38 MAPK inhibition, in growing and quiescent EA.hy926 cells, as defined by cell cycle analysis. The contribution of mitogen- and stress-activated protein kinase (MSK), a downstream target of p38 MAPK, was also investigated. Distinct cellular responses were observed in the two growth states, with p38 MAPK and MSK contributing to the response via different pathways.

## 2. Results

As demonstrated previously, EA.hy926 cells, a macrovascular cell line, that have been maintained in culture for 2 days post-confluence (10 days after subculture) exhibit greatly reduced DNA synthesis compared to cells that have simply reached confluence (8 days after subculture) [11]. These post-confluent cells thus may resemble the quiescent (healthy) state much better than confluent cells. For this reason, growth state comparisons in this study used sub-confluent cells (4 days after subculture) and post-confluent cells (10 days after subculture) as representative populations for the growing and quiescent states, respectively. 

### 2.1. eNOS Was Activated by 20 µM DHA in Both Growth States, but 125 µM DHA Decreased eNOS Activation in Growing Cells

The effects of DHA on eNOS activation in these 2 growth states were assessed systematically by constructing DHA concentration curves at the time point (8 h) at which p38 MAPK showed the maximum differential activation by DHA [12]. Visual inspection of eNOS S1177 phosphorylation 8 h after exposure to different concentrations of DHA (Figure 1a left) revealed that only in quiescent cells did 20 and 40 µM DHA treatments increase the intensity of the p-eNOS band. Statistical analysis of the densitometry values confirmed that 20 µM DHA significantly activated eNOS in quiescent cells (Figure 1b right), but significant activation of eNOS was not found in growing cells (Figure 1b left). A time course of eNOS activation by DHA also established that a relatively low concentration (20 µM) had no effect on either cell viability or proliferation [11]. Quantification of the blots (Figure 1a right) verified that an 8 h treatment with 20 µM DHA activated eNOS in quiescent cells (Figure 1c right) but not in growing cells (Figure 1c left). In growing cells (Figure 1c left), however, eNOS was activated by 20 µM DHA over the entire 2 to 24 h treatment period except at 8 h. Another time course of eNOS activation by a relatively high DHA concentration (125 µM, Figure 1d) demonstrated the opposite effect to that of 20 µM DHA (Figure 1a). Quantification of the relative band intensities confirmed that, instead of activation, 125 µM DHA diminished p-eNOS levels after 8 h in growing cells (Figure 1e left), while p-eNOS remained at a similar level for the entire time course in quiescent cells (Figure 1e left). Taken together, DHA can mediate eNOS activation in endothelial cells in a concentration-, time-, and growth state-dependent manner. While 20 µM DHA can activate eNOS in both growth states at different time points, 125 µM DHA inhibits eNOS activation with prolonged treatment (16 and 24 h) in growing cells.

We next compared, using the same blot, the effect of DHA on eNOS activation and eNOS protein levels 8 h after treatment in both growing and quiescent EA.hy926 cells (Figure 2a). Quantification of the relative band intensities showed that activation of eNOS was not different between the growth states when p-eNOS was normalized against total eNOS (Figure 2b). In contrast, p-eNOS levels were significantly higher in quiescent cells than in growing cells if normalized relative to total protein (Ponceau, Figure 2c). Additionally, a decline in p-eNOS can be detected as a result of 125 µM DHA treatment in both growing and quiescent cells, but the change was statistically significant in quiescent cells only (Figure 2c). Treatment with 125 µM DHA also caused a substantial ~70% decline in total eNOS protein levels (determined by normalization to Ponceau) compared to either 0 or 20 µM DHA treatments (Figure 2d). Furthermore, it was determined that the amount of total eNOS protein is about 1.5-fold higher in quiescent cells than in growing cells (Figure 2d). Therefore, although the relative levels of activated eNOS (p-eNOS/total eNOS) may appear to be similar in the two endothelial cell growth states (Figure 2b), due to differences in total eNOS levels (Figure 2d), the absolute level of activated eNOS (p-eNOS/Ponceau S) in quiescent cells is about twice as much as in growing cells (Figure 2c). The increase in both absolute eNOS activation and total eNOS levels makes the post-confluent (day 10) cells more closely resemble the healthy, quiescent state of endothelial cells.

### 2.2. p38 MAPK and MSK Mediate eNOS Activation by DHA via Distinct Pathways

The involvement of p38 MAPK in eNOS activation by DHA was investigated using a potent p38α/β inhibitor, SB202190 [14]. Under basal conditions without DHA treatment (Figure 3a), active eNOS levels (p-eNOS/total eNOS) were decreased by treatment with the p38 MAPK inhibitor in growing cells but increased in quiescent cells (Figure 3b left). These results suggest that eNOS activation is p38 MAPK-dependent in the growing state but inhibited by p38 MAPK in the quiescent state. This growth state-dependent effect of p38 MAPK on eNOS activation was not observed when p-eNOS levels were normalized to Ponceau (Figure 3b middle). No effect of p38 MAPK on eNOS expression, as defined by total eNOS levels, was observed in the non-DHA-treated cells, although it was again confirmed that quiescent cells had higher eNOS levels than growing cells (Figure 3b right). In response to treatment with 125 µM DHA (Figure 3c), eNOS activation was elevated substantially in growing cells in the presence of the p38 MAPK inhibitor (Figure 3d left). In quiescent cells (Figure 3d right), the inhibitor-dependent elevation in eNOS activation was suppressed by 40 and 125 µM DHA. Additionally, the average intensity of normalized p-eNOS after 20 µM DHA treatment (without inhibitor) was substantially higher (about 2.7-fold) than that of 0 µM DHA treatment in quiescent cells (Figure 3d right), although no statistical significance was found between the two, unlike what was detected in Figure 1b (right). This is likely due to large variations in band intensities among the three independent experiments. In summary, p38 MAPK inhibition had totally opposite effects on eNOS activation in the two endothelial cell growth states, and these effects could be counteracted by DHA treatment in a concentration-dependent manner.

Figure 3e illustrates the effect of p38 MAPK inhibition and DHA on total eNOS levels. In growing cells (Figure 3f left), no effect of the inhibitor was found, while total eNOS levels were elevated by 20 µM DHA. In quiescent cells (Figure 3f right), p38 MAPK inhibition decreased total eNOS levels only in cells treated with 20 µM DHA compared to the respective no inhibitor control. Additionally, 20 µM DHA alone elevated total eNOS levels compared to 0 and 125 µM DHA treatment. Taken together, while at basal level, p38 MAPK signaling may not be required for eNOS expression, the increase in eNOS levels resulting from treatment with 20 µM DHA is dependent upon p38 MAPK but only in quiescent cells.

Since p38 MAPK was required for eNOS activation, it was postulated that MSK, an important direct substrate of p38 MAPK, would also contribute to this process. To examine the involvement of MSK, growing and quiescent EA.hy926 cells were treated with or without SB747651A, a potent MSK inhibitor [15], prior to DHA treatment (Figure 4a). 

In growing cells (Figure 4b left), p-eNOS levels were reduced by MSK inhibition independent of DHA treatment. On the other hand, in quiescent cells (Figure 4b right), eNOS was activated by all 3 concentrations of DHA, while MSK inhibition suppressed this activation for the 20 and 40 µM DHA treatments. At the higher DHA concentration (125 µM), however, eNOS remained activated in quiescent cells compared to 0 and 20 µM DHA plus MSK inhibitor treatment. Therefore, eNOS activation is dependent upon MSK in both growth states, with or without DHA, except in quiescent cells exposed to a high concentration (125 µM) of DHA. Based on these results, it is obvious that the response of eNOS activation to p38 MAPK inhibition (Figure 3b,d) and MSK inhibition (Figure 4b) in conjunction with DHA treatment was not the same. Therefore, these two kinases operate independently when signalling to eNOS in response to DHA.

With respect to total eNOS levels (Figure 4c), no effect of MSK inhibition or DHA treatment was found for growing cells (Figure 4d left), but in quiescent cells, the MSK inhibitor reduced total eNOS protein levels in the absence of DHA compared to its without-inhibition control (Figure 4d right). In addition, 125 µM DHA significantly decreased total eNOS levels compared to the 0 µM DHA condition, both with and without MSK inhibition. Overall, these results indicate that MSK is needed for basal eNOS expression in quiescent cells only.

### 2.3. Growth State Determines Whether DHA Induces Apoptosis of Endothelial Cells 

With the finding that the day 10 EA.hy926 cells had more activated and total eNOS (Figure 2), cell cycle analysis, based on propidium iodide (PI) staining (Figure 5a), was used to compare the two endothelial growth states to better examine the resemblance of day 10 cells to the healthy, quiescent state present in vivo. The experimental data showed that at day 10 there were many more cells in G0/G1 (71.29% ± 1.55 vs. 88.33% ± 0.49, Figure 5b) and fewer cells in S (2.26% ± 0.13 vs. 11.45% ± 0.69, Figure 5c) and G2/M (5.77% ± 0.20 vs. 10.68% ± 0.91, Figure 5d) compared to day 4. Furthermore, the results show that DHA has no effect on the cell cycle of endothelial cells within the 8 h treatment period. These data establish that the majority of cells at day 10 were arrested in the G0/G1 phase, supporting the notion that the day 10 cells resemble the healthy, quiescent state. 

Our lab has previously reported that DHA induced apoptosis in confluent (day 8) EA.hy926 cells via p38 MAPK-activated caspase-3, -8 and -9 [12]. Here, we further examined this apoptosis-inducing effect of DHA in the two growth states by comparing the number of cells in sub G0/G1, which primarily consists of apoptotic cells [16]. The percentage of apoptotic cells was not different in growing and quiescent cells in the absence of DHA (Figure 5e). However, the presence of DHA altered the number of cells in apoptosis in a growth state-dependent manner. Treatment with 125 µM DHA increased the percentage of growing cells in sub G0/G1 compared to untreated (0 µM DHA) cells (4.64% ± 0.85 vs. 1.31% ± 0.14, Figure 5e), matching our previous report that DHA at this concentration reduced endothelial cell viability [11]. On the other hand, quiescent cells had a lower percentage of sub G0/G1 cells when treated with 20 µM (0.94% ± 0.12 vs. 2.96% ± 0.90) and 125 µM DHA (2.63% ± 0.55 vs. 4.64% ± 0.85) versus growing cells exposed to the same concentrations of DHA (Figure 5e). Taken together, growing endothelial cells are sensitive to induction of apoptosis by DHA, whereas quiescent cells are more resistant to the actions of DHA with respect to apoptosis.

## 3. Discussion

This is the first study to systematically evaluate the effects of DHA on both eNOS activation and total protein levels in growing and quiescent human endothelial cells, which are representative of the dysfunctional and healthy states in vivo, respectively. There were several novel findings in this study, which used DHA concentrations that were well within the plasma DHA concentration range (7.2 to more than 400 µM) achievable by diet and/or supplementation in the general population [17,18]. In quiescent cells, a low concentration (20 µM) of DHA activated eNOS at 8 h into the time course, whereas treatment of growing cells with the same concentration of DHA activated eNOS over 2 distinct time periods (2–4 h and 16–24 h). On the other hand, a high (125 µM) concentration of DHA for a prolonged treatment time (≥8 h) lowered the level of activated eNOS in growing but not quiescent cells. It was also found, for the first time, that the ability of DHA to decrease or increase total eNOS protein levels was dependent upon the DHA concentration. This effect of DHA was observed with cells in both growth states, although quiescent endothelial cells had higher basal levels of eNOS than growing cells. Our attempts to elucidate possible mechanisms behind this concentration- and growth state-dependent activation of eNOS by DHA revealed that p38 MAPK differentially affected eNOS activation in growing and quiescent endothelial cells, and its effects could be counteracted by DHA treatment in a concentration-dependent manner. MSK, an important p38 MAPK substrate, was also found to mediate eNOS activation, and this observation has not been previously described. In addition to eNOS phospho-activation, p38 MAPK and MSK also modulated eNOS expression in terms of its total protein levels. However, the signal was not transduced through the p38 MAPK/MSK pathway. Instead, the two kinases acted independently with respect to eNOS activation and its expression in response to DHA. 

In this study, DHA was found to differentially activate eNOS, as measured by the level of Ser-1177 phosphorylation, and the response to DHA was dependent on time, concentration and growth state (Figure 1 and Figure 2). Although 20 µM DHA has been reported to elevate p-eNOS Ser-1177 levels in endothelial cells 24 h [19] and 48 h [20] after treatment, matching our finding as shown in Figure 1c, few papers have comprehensively evaluated the effects of treatment time, DHA concentration and endothelial cell growth state on eNOS activation. The importance and physiological relevance of eNOS phosphorylation at Ser-1177 has been well established in vivo using transgenic mice [21,22,23], thus confirming its utility as a marker for eNOS activation and an indirect indicator of endothelial health. This level of in vivo evidence has not been obtained for other eNOS phosphorylation sites [2]. Still, the effect of DHA on eNOS activation would be more conclusive if the phosphorylation data (Figure 1 and Figure 2) agreed with the results from other eNOS activity assays, such as the citrulline assay. Such data are available in the literature where Li et al. [24] found that 25 µM DHA significantly increased by almost 2-fold the catalytic activity of eNOS after 24 h treatment, which matches our data for 20 µM DHA in growing cells (Figure 1c left). Although we did not find significant activation of eNOS by DHA at concentrations lower than 20 µM for the time point tested (8 h), longer treatment times of 24 h with 3 and 10 µM DHA [9,25] as well as 48 h with 10 µM DHA [26] were found to increase NO production. Chronic treatment for 7 days with even lower DHA concentrations (5 and 50 nM) increased both eNOS phosphorylation and NO production [27]. The cell models used in all of these studies were either primary endothelial cells or immortalized cell lines such as EA.hy926 but grown in the absence of an extracellular matrix coating. The presence of a matrix is critical, since in its absence, the cells do not become quiescent even when they are confluent [11]. Therefore, limited data are available for the induction of eNOS activation by DHA in quiescent cells that more accurately resemble the natural state of the endothelium. What is clear, however, is that eNOS exhibits distinct phosphorylation patterns in response to DHA in growing and quiescent cells. While the evidence for low concentrations of DHA activating eNOS is strong, data showing higher DHA concentrations (>100 µM) that also influence eNOS activation are scarce. Only Chao et al. [28] reported that treatment with 100 µM DHA for 8 h may decrease VEGF-induced eNOS phosphorylation, thus reducing cell migration. However, they did not test the effect of DHA alone. This lack of data may be due to DHA triggering apoptosis at higher concentrations as demonstrated in our cell cycle results (Figure 5e). In addition, our data showed that at those higher concentrations (125 and 150 µM), DHA has a limited effect on eNOS activation in comparison to 20 µM (Figure 1b).

In addition to the effects on eNOS activation, DHA was also found to either decrease total eNOS levels (Figure 2d) or increase total eNOS levels (Figure 3f) depending upon the concentration to which the cells were exposed. While treatment with 20 µM DHA elevated total eNOS levels under the experimental conditions (Figure 3f), 125 µM DHA consistently decreased total eNOS levels (Figure 2d and Figure 5d). Furthermore, the combined results of Figure 2d, Figure 4f and Figure 5d suggest that the decline in eNOS expression that occurs in response to 125 µM DHA may be more profound in quiescent cells than in growing cells. Unlike what was found in the current study, specifically that DHA may differentially affect the total eNOS levels as just discussed, some publications reported no effect of DHA on total eNOS protein [20,27], while others found that DHA impeded the decrease in *NOS3* mRNA levels by TNF-α or palmitic acid [29,30]. This discrepancy may be the result of differences in DHA concentrations used, as indicated in this study. Those studies that found no effect of DHA used a maximum of 12 µM, while the others used 30–50 µM DHA. As discovered in our study, 20 µM DHA may elevate total eNOS levels in both growth states (Figure 3f), consistent with the beneficial effects found by Yamagata et al. [16] and Novinbahador et al. [17]. On the other hand, high concentrations of DHA, reaching 125 µM, may produce deleterious effects by reducing eNOS protein levels (Figure 2d and Figure 5d right). These concentration-dependent actions of DHA on total eNOS levels, which are similar to those observed for eNOS activation and cell cycle profiling, support the need for more research to define the acceptable dose limit, which could instigate refinement of DHA intake recommendations.

The results shown in Figure 3 clearly support our hypothesis that p38 MAPK is involved in the activation of eNOS by DHA. What is more intriguing is our novel finding that modulation of eNOS activity by p38 MAPK may be dictated by cell phenotype: basal level eNOS activation requires active p38 MAPK in growing cells but is inhibited by this kinase when cells are in the quiescent state (Figure 3b left). This growth state-dependent action of p38 MAPK may explain the previously reported controversies regarding the relationship between p38 MAPK and eNOS activation [7,31]. Upon inclusion of DHA treatment, the situation becomes more complex. In growing cells, 125 µM DHA was reported to activate PI3K/Akt signaling [32] leading to eNOS phosphorylation, but this did not occur at lower concentrations [33]. These reports agree with our results showing that only 125 µM DHA significantly activates eNOS when p38α/β MAPK is inhibited (Figure 3d left). Therefore, DHA may induce eNOS phosphorylation via other pathways such as PI3K/Akt in growing cells, which can then be inhibited by p38 MAPK. In quiescent cells, on the other hand, higher concentrations of DHA interfered with the eNOS activating effect of p38 MAPK inhibition and downregulated p-eNOS levels that were otherwise elevated by the inhibitor (Figure 3d right). This indicates that there should be a signaling pathway which inhibits eNOS activation that can be stimulated by high concentrations of DHA in quiescent endothelial cells. However, little information on this point can be found in the literature; thus, further research examining the mechanisms affecting both eNOS activation and inhibition is required. In this context, we examined intermediates of the p38 MAPK signaling pathway and found, for the first time, that MSK is also required for eNOS activation in both growing and quiescent endothelial cells (Figure 4b). Although MSK is an important p38 MAPK substrate, the responses of eNOS activation to p38 MAPK inhibition and to MSK inhibition were fundamentally different (Figure 3d vs. Figure 4b), indicating that the effect of MSK on eNOS activation is unlikely to be coupled to p38 MAPK. MSK can also be activated by ERK1/2 [34], which was reported to be activated by DHA in both growth states [35]. Therefore, it is possible that DHA may act via ERK1/2 to activate MSK, which then mediates eNOS phosphorylation in both growth states.

Apart from eNOS activation, total eNOS protein levels may also be regulated by p38 MAPK (Figure 3f) and MSK (Figure 4d). Again, changes in total eNOS levels in response to p38 MAPK and MSK inhibition indicate that these two kinases act independently. p38 MAPK was reported to downregulate *NOS3* promoter activity, thus lowering *NOS3* gene expression [31,36]. We also observed a subtle increase in eNOS protein levels in p38 MAPK-inhibited cells compared to their respective controls, but statistical significance was not reached (Figure 3b,f). However, this relationship may not hold in quiescent cells (Figure 3f right), especially with 20 µM DHA treatment: for cells in this state, p38 MAPK signalling was required for the increase in eNOS protein levels by DHA. This response might be linked to CREB signalling at the transcriptional level, since CREB can be activated via the p38 MAPK/MSK pathway and operates through AP-1 to promote *NOS3* gene transcription [37]. Additionally, in quiescent cells only, MSK was essential for basal level expression of eNOS (Figure 4d). Therefore, one possible mechanism by which MSK modulates eNOS expression may be via activation of transcription factors, such as CREB [37] and/or NF-κB [38], which are linked to *NOS3* gene transcription.

The cell cycle results support the view that the post-confluent (day 10) cells in our model resembled the quiescent state, with many more cells in G0/G1 (Figure 5b) and fewer in both S (Figure 5c) and G2/M (Figure 5d) compared to the sub-confluent (day 4) growing state. The very low percentage of cells in S phase for the post-confluent (day 10) state is also in agreement with the DNA synthesis results obtained previously with this cell model of EA.hy926 endothelial cells cultured on Matrigel^®^-coated plates [11]. In addition, post-confluent day 10 cells had about twice the amount of activated-eNOS (Figure 2c) and 50% higher total eNOS protein levels (Figure 2d) compared to day 4 cells; this further supports the similarity between day 10 cells and the healthy, quiescent state in vivo, with more eNOS available for NO production compared to the proliferative, dysfunctional state represented by the growing cells. These differences in eNOS activation and eNOS levels between the two growth states of endothelial cells have not been reported before; this is an important observation given the role of eNOS for maintaining vascular homeostasis in healthy endothelial cells.

The cell cycle profile of endothelial cells was altered by DHA, with 125 µM DHA increasing the number of apoptotic cells (sub G0/G1) in the growing state and 20 µM DHA decreasing the number of cells in sub G0/G1 in the quiescent state (Figure 5e). The effect of DHA on apoptosis and cell cycle regulation is well-established, especially in cancer cells where DHA has potential therapeutic uses [39]. Kim et al. [40] reported that DHA induced apoptosis in growing human umbilical vein endothelial cells (HUVEC) but not confluent cells, in agreement with our results in growing cells treated with 125 µM DHA and the attenuated apoptosis of quiescent cells treated with 125 µM DHA (Figure 5e). However, the previous report from our lab found that DHA induced apoptosis in confluent (day 8) EA.hy926 cells grown on Matrigel^®^ [12]. The discrepancy between the two papers may lie in the different concentrations used: one paper used 20 and 40 µM DHA [40], whereas the other employed DHA at 125 µM [12]. Furthermore, compared to the finding in our previous report [12] that 125 µM DHA induced apoptosis in confluent (day 8) cells, this high concentration of DHA did not induce apoptosis in quiescent (day 10) cells (Figure 5e). Therefore, the effects of DHA on endothelial cell apoptosis depend on concentration and growth state, with quiescent cells being resistant to the apoptosis-inducing effect of DHA compared to both growing and confluent cells. Consequently, we propose that this growth state-dependent effect of DHA on apoptosis could be caused by p38 MAPK-activated caspases as demonstrated by our lab previously [12].

Figure 6 illustrates several plausible mechanisms by which DHA could modulate eNOS activation and/or expression in a concentration- and growth state-dependent manner. The divergent cell signaling pathways in the two growth states may be one of the explanations for the growth-state-dependent effects of DHA, as discussed above. Apart from acting through different signaling pathways, DHA could also be metabolized into a wide range of bioactive lipid mediators, such as oxylipins. Different oxylipins were found to have different pharmacokinetics and final concentrations after DHA supplementation [41,42], and these DHA metabolites can trigger different signaling pathways depending on their variety and concentration [43]. In addition, there are many possible receptors of DHA in endothelial cells, such as GPR120 (FFAR4) [28] and PPAR [44]. Each of those receptors may have different affinities for DHA and thus may convey different intracellular signals due to their ability to respond to different concentrations of DHA and/or its metabolites [45]. It is also possible that certain receptors are expressed in one growth state but not the other. Therefore, different effects of DHA were observed at different concentrations in different cell growth states. Further research is needed to elucidate in greater detail the mechanisms that underpin these complex DHA-induced changes in eNOS activation and expression. 

## 4. Materials and Methods

General laboratory chemicals were obtained from Fisher Scientific International, Inc. (Hampton, NH, USA) or Sigma-Aldrich (St. Louis, MO, USA).

### 4.1. Cell Culture and Treatment

EA.hy926 cells (Catalog #: CRL-2922, American Type Culture Collection, Manassas, VA, USA) were cultured to growing (sub-confluent) and quiescent (2 days post-confluent) states on Matrigel^®^ (Catalog #: 356231, Corning^®^, Corning, NY, USA) as described previously [11]. The growing state is defined as sub-confluent cells at day 4 after seeding, whereas the post-confluent condition at day 10 is considered the quiescent state. Cells were treated with various concentrations of DHA for different time durations as indicated in the figures, with or without addition of the indicated inhibitor (SB202190 or SB747651A, Catalog #: 1264 and 4630, respectively, Tocris, Bristol, UK) 30 min prior to the DHA treatment. DHA (Catalog #: 90310, Cayman Chemical, Ann Arbor, MI, USA) was initially dissolved in ethanol and then conjugated to fatty-acid-free bovine serum albumin (BSA, Catalog #: 10775835001, Roche, Basel, Switzerland) in phosphate-buffered saline (PBS) for cell delivery [11]. An equivalent amount of ethanol in 5% BSA-PBS was used as vehicle control, which is referred to as Ctl or 0 µM DHA in the figures and figure legends.

### 4.2. Cell Cycle Analysis

Treated EA.hy926 cells on 6-well plates were harvested with trypsin (Catalog #: 25200114, Gibco™, Waltham, MA, USA) for cell cycle analysis using flow cytometry as previously described [46]. Briefly, after aspirating culture media, trypsinized cells were centrifuged at 750× *g* for 5 min and then washed once with PBS prior to ethanol fixation (5 mL of ice-cold 75% ethanol in PBS per well for 2 h). The ethanol was removed via centrifugation (300× *g* for 5 min), and the cell pellets were washed once with cold PBS before being resuspended in 0.5 mL PI solution (50 μg/mL PI (Catalog #: P4864, Sigma-Aldrich, St. Louis, MO, USA), 4 mM Na citrate, 0.1% Triton X-100, 50 μg/mL RNase A (Catalog #: T3018L, New England Biolabs, Ipswich, MA, USA), pH adjusted with NaOH to 7.8). The cells were incubated in the dark for 10 min at 37 °C, and 50 μL of 1.38 M NaCl was added to the 0.5 mL solution in each tube immediately after incubation. The samples were carefully pipetted to achieve a single cell suspension and then analyzed by flow cytometry (CytoFlex LX Digital Flow Cytometry Analyzer 4 Laser System (Beckman Coulter, Brea, CA, USA)) at the University of Manitoba Flow Cytometry Core Facility. On average, >15,000 gated events were analyzed to obtain the percentage of cells in sub G0/G1, G0/G1, S and G2/M phases of cell cycle based on the mean fluorescent intensity of PI.

### 4.3. Western Blotting

Treated cells were harvested with 2× sample buffer (125 mM Tris-HCl pH 6.8, 2% sodium dodecyl sulphate (SDS), and 20% glycerol) and then subjected to a brief sonication and centrifugation prior to protein quantification with a BCA assay kit (Catalog #: 23225, Pierce, Thermo Fisher Scientific, Inc., Waltham, MA, USA). Protein (15 µg per sample) was heated in the presence of β-mercaptoethanol, then separated by SDS-PAGE, and transferred onto a PVDF membrane (Catalog #: 03010040001, Roche, Basel, Switzerland). Triple Wide Mini electrophoresis (Catalog #: MGV-202-33) and blotting (Catalog #: EBU-302) systems from C.B.S. Scientific (San Diego, CA, USA) were used to run all the replicates from the same experiment together on a single gel/blot. The blots were blocked in 3% BSA-TBST for at least 1 h at room temperature, incubated overnight at 4 °C with the respective primary antibodies (diluted in 3% BSA-Tris-buffered saline with Tween-20 (TBST))—anti-phospho-eNOS Ser1177 (1:1000, Catalog #: 9571, Cell Signaling Technology, Danvers, MA, USA) and anti-eNOS (1:1000, Catalog #: 9572, Cell Signaling Technology, Danvers, MA, USA)—and then probed with secondary antibody (1:10,000 in 1% BSA-TBST) at room temperature for 1 h. For each data set, the blot was incubated for 15 min at ambient temperature with shaking in stripping buffer (0.2 M glycine-HCl pH 2.0, 1% SDS, supplemented with 0.8% β-mercaptoethanol immediately before use) to selectively remove the anti-p-eNOS antibody, after which the blot was blocked with 3% BSA-TBST for 1 h prior to re-probing with anti-eNOS antibody, as described above. Ponceau S (Catalog #: 97063-650, VWR, Radnor, PA, USA) staining was used to assess total protein load for normalization [47]. Blot images were captured with a ChemiDoc Imager (Bio-Rad, Hercules, CA, USA) and quantified using Image Lab software version 6.0.1 (Bio-Rad, Hercules, CA, USA).

### 4.4. Statistical Analysis

All data were analysed using IBM SPSS statistics version 27 (IBM Corp., Armonk, NY, USA) and plotted as means ± standard error of the mean (SEM) using GraphPad Prism version 9.0 (GraphPad software, San Diego, CA, USA). Outliers were identified as those outside the range of the mean ± 2.5 standard deviations of the dataset and were removed prior to analysis. For the DHA concentration curve and time course datasets, one-way ANOVA followed by post hoc testing with Duncan’s multiple range test (for homogenous data sets) or Dunnett’s test (for non-homogenous data sets) was used. If the dataset was not normally distributed, log transformation was performed before subjecting it to the analysis. For data studying the combined effects (interactions) of DHA × inhibitor or DHA × growth state, two-way ANOVA (main effects and interaction) followed by pair-wise mean comparison with Bonferroni correction was employed. Statistical significance was set at *p* < 0.05, except that an interaction in a two-way ANOVA was considered significant at *p* < 0.1. A significant interaction term, for example DHA × inhibitor, indicated that the effect(s) of an inhibitor varied with the DHA concentration. 

## 5. Conclusions

Proper functioning of endothelial cells is critical for vascular health, and this includes maintaining proper cell proliferation status and enhancing NO bioavailability via eNOS. The present study has demonstrated that a low concentration of DHA (20 µM) may be beneficial to vascular health by activating eNOS, possibly increasing its total protein levels, and suppressing apoptosis when endothelial cells are in the quiescent state. On the other hand, a high concentration of DHA (125 µM) could be deleterious by blocking eNOS activation, lowering total eNOS levels, especially in quiescent cells, and increasing apoptosis in growing cells. p38 MAPK was found to up- or down-regulate eNOS activation depending on the growth state, and DHA can counteract these effects. MSK was also required for eNOS activation in both growth states and for eNOS expression in the quiescent state, but this effect was independent of DHA and was not linked to p38 MAPK. The results of this study emphasize the importance of growth state in the endothelial cell response to DHA. As a result of these findings, future research examining the relationship between DHA and an individual’s health status is needed in order to help refine recommendations for DHA consumption via either diet or supplementation. This is important because health status could have considerable bearing on whether the expected benefits of this n3 PUFA with respect to its potential atheroprotective effects are actually realized. Additionally, the differences in cell signaling in the two endothelial growth states may serve as the initial point for exploring possible therapeutic targets and/or diagnostic markers of endothelial cell function and atherosclerosis. 

## Figures and Tables

**Figure 1 ijms-24-08346-f001:**
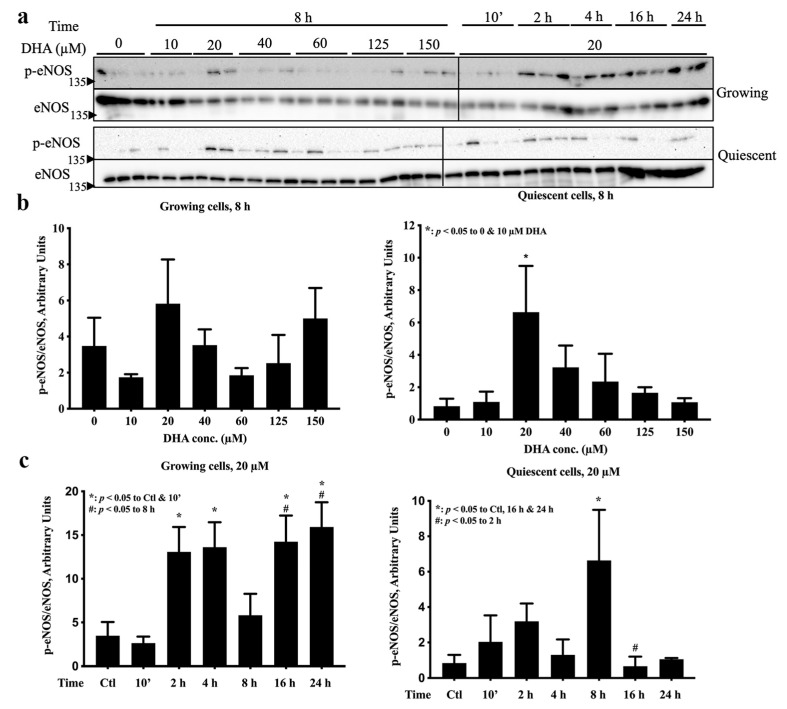
Activation of eNOS by DHA in endothelial cells is growth-state-, concentration-, and time-dependent. Growing and quiescent EA.hy926 cells were treated with various concentrations of DHA for different durations, and Western blotting was used to measure p-eNOS (Ser-1177) and total eNOS. (**a**) Western blots of DHA concentration titrations for 8 h (left half) and time course with 20 µM DHA (right half) in growing and quiescent cells. Graphic representations of the quantified band intensities for 8 h DHA concentration titrations are in panel (**b**), and the time courses for 20 µM DHA are in panel (**c**). (**d**) Western blots of time courses for 125 µM DHA in growing and quiescent cells. (**e**) Graphic representations of the quantified band intensities for time courses with 125 µM DHA treatments. In panels (**a**,**d**), the blots show 3 biological replicates per treatment. In panels (**b**,**c**,**e**), the data are presented as means ± SEM (n = 3); different symbols denote statistical significance (*p* < 0.05) between different treatment conditions as indicated within each graph, based on one-way ANOVA followed by post hoc testing with Duncan’s multiple range test (for homogenous data sets) or Dunnett’s test (for non-homogenous data sets). Ctl: vehicle control with 0 μM DHA treatment. The band intensity data from panel (**a**) for Ctl and for 20 µM DHA at 8 h were used for both the DHA concentration titration at 8 h (**b**) and for the time courses with 20 μM DHA (**c**), since the samples were identical for these conditions and included on the same blot.

**Figure 2 ijms-24-08346-f002:**
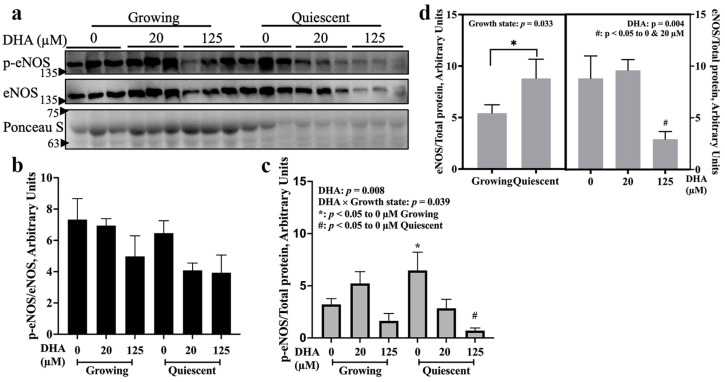
125 µM DHA reduced both absolute p-eNOS levels and total eNOS protein in endothelial cells. Growing and quiescent EA.hy926 cells were treated with 20 or 125 µM DHA for 8 h, and the cell extracts were analysed by Western blotting for p-eNOS (Ser-1177) and total eNOS. Representative blots with 3 biological replicates are shown in panel (**a**). Graphical representations of the quantified band intensities for (**b**) p-eNOS normalized to total eNOS, (**c**) p-eNOS normalized to total protein load (Ponceau S staining) and (**d**) total eNOS normalized to total protein load, show means ± SEM (n = 3 for panels (**b**,**c**); for panel (**d**), n = 9 for the main effect of growth state, and n = 6 for the main effect of DHA concentration). Each graph provides *p* values for significant main effects (*p* < 0.05; DHA or growth state), significant interactions (*p* < 0.1; DHA × growth state) from a two-way ANOVA, and the different symbols denote significant (*p* < 0.05) differences between treatment conditions based on pair-wise mean comparisons with Bonferroni correction.

**Figure 3 ijms-24-08346-f003:**
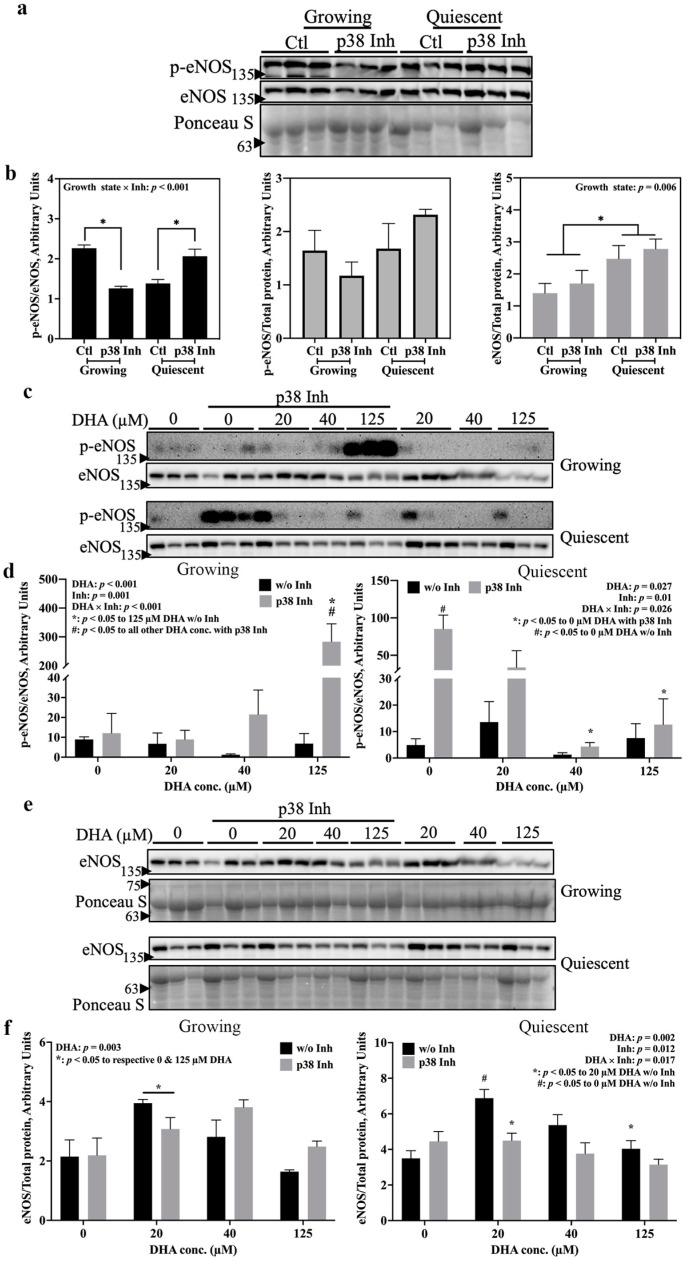
Activation of eNOS, without or with DHA, in endothelial cells is mediated by p38 MAPK in a growth state-dependent manner. Growing and quiescent EA.hy926 cells were treated for 8 h with the p38 MAPK inhibitor, SB202190 (1 μM), independently or in combination with various concentrations of DHA. Western blotting was used to measure p-eNOS (Ser-1177), total eNOS and total protein load (Ponceau S staining). (**a**) Western blots of growing and quiescent EA.hy926 cells treated with or without the p38 MAPK inhibitor in the absence of DHA; graphic representations of the quantified band intensities in those blots are shown in panel (**b**). Western blots of (**c**) eNOS activation and (**e**) total eNOS levels in growing and quiescent EA.hy926 cells that were exposed to various concentrations of DHA with or without the p38 MAPK inhibitor. Graphic representations of the quantified band intensities of (**d**) eNOS activation and (**f**) total eNOS levels are shown beneath the respective blots. In panels (**a**,**c**,**e**), the blots show 3 biological replicates per treatment, except the 40 µM DHA conditions which have 2 biological replicates per treatment. In panels (**b**,**d**,**f**), the data are presented as means ± SEM (n = 3; except n = 2 for the p38 Inh treatment in quiescent cells in (**b**) middle and right, and 40 µM DHA treatments in (**d**,**f**)). Each graph provides *p* values for significant main effects (*p* < 0.05; growth state or Inh in (**b**) and DHA or Inh in (**b**,**c**)) and for significant interactions (*p* < 0.1; growth state × Inh in (**a**) and DHA × Inh in (**b**,**c**)) from a two-way ANOVA. The different symbols denote significant (*p* < 0.05) differences between treatment conditions based on pair-wise mean comparisons with Bonferroni correction. Ctl: vehicle control; Inh: inhibitor.

**Figure 4 ijms-24-08346-f004:**
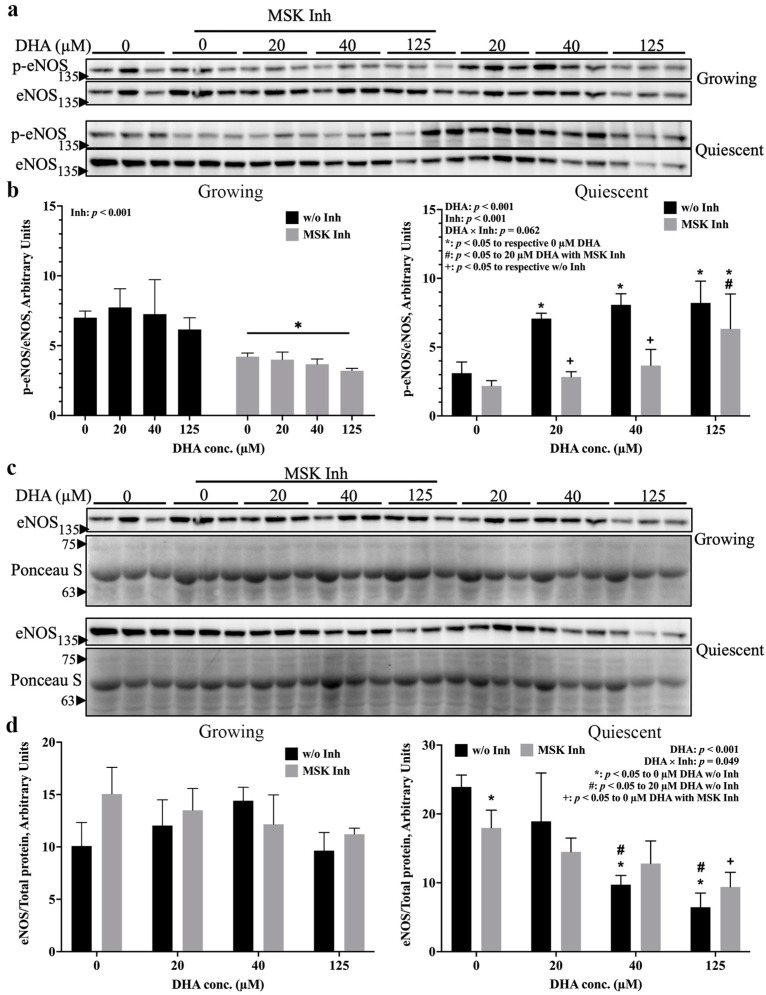
MSK signaling is required for eNOS activation in both growth states and for increasing eNOS levels in quiescent endothelial cells. Growing and quiescent EA.hy926 cells were exposed to various concentrations of DHA for 8 h with or without the MSK inhibitor, SB747651A (5 μM). Western blotting was used to measure (**a**,**b**) eNOS activation (p-eNOS Ser-1177) and (**c**,**d**) total eNOS levels; the blots (**a**,**c**) show three biological replicates. Graphic representations of the quantified band intensities of (**b**) eNOS activation and (**d**) total eNOS levels show means ± SEM (n = 3). Each graph provides *p* values for significant main effects (*p* < 0.05; DHA or Inh), and significant interactions (*p* < 0.1; DHA × Inh) from a two-way ANOVA. The different symbols denote significant (*p* < 0.05) differences between treatment conditions based on pair-wise mean comparisons with Bonferroni correction. Inh: inhibitor.

**Figure 5 ijms-24-08346-f005:**
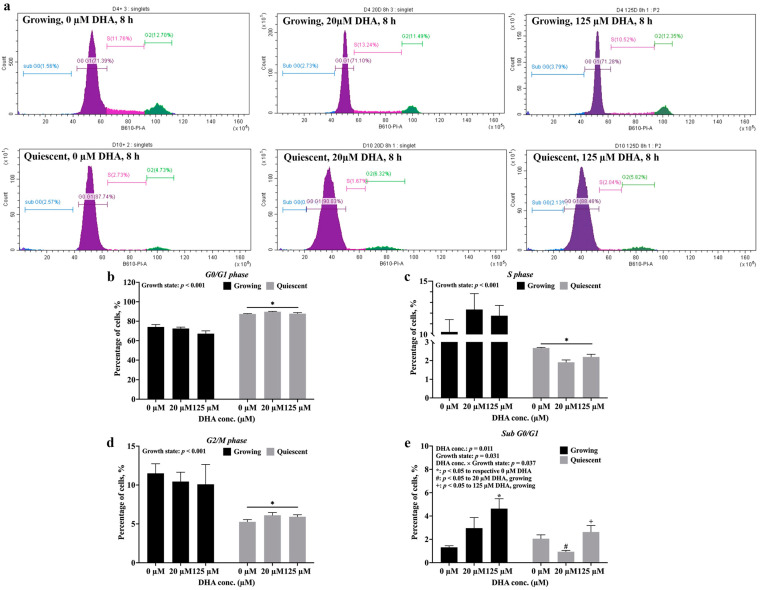
Quiescent endothelial cells are more resistant to the induction of apoptosis by DHA than growing cells. Growing and quiescent EA.hy926 cells were treated with 0, 20 and 125 μM DHA for 8 h before fixation with ethanol, PI staining and analysis by flow cytometry. (**a**) Representative histograms for growing (top) and quiescent (bottom) EA.hy926 cells. Computed percentages of cells are graphically presented for (**b**) G0/G1, (**c**) S, (**d**) G2/M and (**e**) subG0/G1 phases as means ± SEM (n = 3; except n = 2 for 125 µM DHA treatment in growing cells for sub G0/G1 measurements (**e**)). Each graph provides *p* values for significant main effects (*p* < 0.05; DHA or growth state) and significant interactions (*p* < 0.1; DHA × growth state) from a two-way ANOVA. The different symbols denote significant (*p* < 0.05) differences between treatment conditions based on pair-wise mean comparisons with Bonferroni correction.

**Figure 6 ijms-24-08346-f006:**
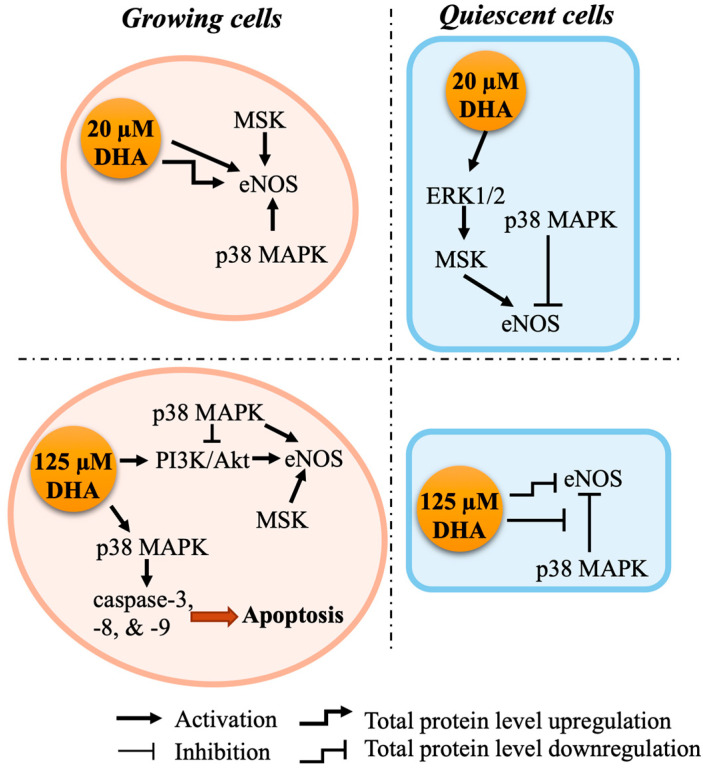
Proposed mechanisms of action of DHA in growing and quiescent endothelial cells at low (20 µM) and high (125 µM) DHA concentrations.

## Data Availability

The study data are available upon reasonable request.

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
