# Peer review of "Growth State-Dependent Activation of eNOS in Response to DHA: Involvement of p38 MAPK"

_ijms, 2023, doi:10.3390/ijms24098346_

Round 1

Reviewer 1 Report

The authors tested the effects of DHA on both quiescent and growing endothelial cells, and found that DHA regulates p38 MAPK and eNOS activity in a growth state-dependent manner. It is an interesting topic, but there are several major concerns regarding to the study design and novelty.

1. The major aim of the study is to test the effects of DHA on eNOS activity in quiescent and growing endothelial cells. The authors used eNOS phosphorylation to indicate its activity. It should be noted that eNOS phosphorylation is different from eNOS activity. NO production, and other indices should be provided.

2. The authors previously reported that DHA differentially activates p38 MAPK in growing and quiescent endothelial cells. The study only extends the previously findings, which shows that DHA regulates eNOS activity in dependent of p38 MAPK pathway. Thus, the study lacks novelty. I do not think the study provides enough new information here.

3. The methods used here for establishment of growing and quiescent cells are questionable. Whether the methods used here were reliable needs further evidence.

4. There was no internal reference in most of the WB experiments. In Figure 2a, the authors used Ponceau S as internal reference. It is not accepted.

5. To test the role of p38 MAPK and MSK, the authors only used inhibitors in the experiments. The evidence provided here is not solid enough.

6. The study was almost observational. The molecular mechanisms should be well explored.

7. The authors tested the cell resistance to apoptosis induced by DHA in Figure 5. I do not think these experiments were highly associated with the aim of the study. It is the other question which should be presented in the other paper.

Author Response

We would like to thank the Reviewer for the comprehensive assessment of our manuscript and the thoughtful comments made to help improve it. We have provided further supporting evidence in the replies and hope that they will be satisfactory. A detailed summary of these replies and the manuscript file indicating where the changes have been made has been provided.

Reviewer 2 Report

Following on from a previous study, the authors present an interesting study in which they further explore the effects of docosahexaenoic acid (DHA) on p38 MAPK signalling in the endothelial cells. Briefly, the authors utilise two models of endothelial cells; healthy and growing, and quiescent and non-growing, to examine whether DHA similar or differential effects in both cellular states. Both cell models saw eNOS respond to DHA treatment, albeit with different effects at different concentrations in each model respectively in certain instances. Utilising inhibitors of the p38 MAPK pathway, it was determined MSK in particular had an influential role on DHA signalling, but once again, with differential effects depending on the model. Ultimately, this study shares insights into the mechanism of DHA and how it may influence endothelial health dependent on the health status of the cells themselves.  

In reviewing the manuscript, I made a number of observations. The following should be addressed by the authors when preparing a suitable revision.

1.       Were any vehicle controls included as part of this study to determine the net effect of DHA on cell behaviours?

2.       While there are some data on the cell cycle, were any viability studies conducted on the DHA compound and other aspects of the study to determine the effect on cell health/number?

3.       More details on how the flow cytometry was conducted would be useful. For example, were floating cells included as part of the analysis? How many events were counted/analysed? Etc.

4.       How was the concentration of DHA selected for this study?

5.       Were separate blots run for each western blot i.e. for eNOS and p-eNOS, or were the blots stripped and reprobed for each molecular target?

6.       The authors seem to indicate that ponceau staining was used to normalise the loading of the blots, however, the use of ponceau in this context is questionable. Why was another target such as GAPDH or beta actin not used?

7.       The diagrams in Figure 6, while good in idea, could be improved somewhat in the design to make the overall messages clearer.

Author Response

We would like to thank the Reviewer for the comprehensive assessment of our manuscript and the thoughtful comments made to help improve it. We hope that our replies and modifications made accordingly will be satisfactory. A detailed summary of these replies and the manuscript file indicating where the changes have been made has been provided.

Round 2

Reviewer 1 Report

I do not think the authors have addressed my concerns well.

Reviewer 2 Report

The authors have suitably addressed my comments and the manuscript is much improved.